# MODEL-CENTRIC DATA MANIFOLD:
# THE DATA THROUGH THE EYES OF THE MODEL

## ABSTRACT

We discover that deep ReLU neural network classifiers can see a low-dimensional Riemannian manifold structure on data. Such structure comes via the *local data matrix*, a variation of the Fisher information matrix, where the role of the model parameters is taken by the data variables. We obtain a foliation of the data domain and we show that the dataset on which the model is trained lies on a leaf, the *data leaf*, whose dimension is bounded by the number of classification labels. We validate our results with some experiments with the MNIST dataset: paths on the data leaf connect valid images, while other leaves cover noisy images.

## 1 INTRODUCTION

In machine learning, models are categorized as discriminative models or generative models. From its inception, deep learning has focused on classification and discriminative models (Krizhevsky et al., 2012; Hinton et al., 2012; Collobert et al., 2011). Another perspective came with the construction of generative models based on neural networks (Kingma & Welling, 2014; Goodfellow et al., 2014; Van den Oord et al., 2016; Kingma & Dhariwal, 2018). Both kinds of models give us information about the data and the similarity between examples. In particular, generative models introduce a geometric structure on generated data. Such models transform a random low-dimensional vector to an example sampled from a probability distribution approximating the one of the training dataset. As proved by Arjovsky & Bottou (2017), generated data lie on a countable union of manifolds. This fact supports the human intuition that data have a low-dimensional manifold structure, but in generative models the dimension of such a manifold is usually a hyper-parameter fixed by the experimenter. A recent algorithm by Peebles et al. (2020) provides a way to find an approximation of the number of dimensions of the data manifold, deactivating irrelevant dimensions in a GAN.

Similarly, here we try to understand if a discriminative model can be used to detect a manifold structure on the space containing data and to provide tools to navigate this manifold. The implicit definition of such a manifold and the possibility to trace paths between points on the manifold can open many possible applications. In particular, we could use paths to define a system of coordinates on the manifold (more specifically on a chart of the manifold). Such coordinates would immediately give us a low-dimensional parametrization of our data, allowing us to do dimensionality reduction.

In supervised learning, a model is trained on a labeled dataset to identify the correct label on unseen data. A trained neural network classifier builds a hierarchy of representations that encodes increasingly complex features of the input data (Olah et al., 2017). Through the representation function, a distance (e.g. euclidean or cosine) on the representation space of a layer endows input data with a distance. This pyramid of distances on examples is increasingly class-aware: the deeper is the layer, the better the metric reflects the similarity of data according to the task at hand. This observation suggests that the model is implicitly organizing the data according to a suitable structure.

Unfortunately, these intermediate representations and metrics are insufficient to understand the geometric structure of data. First of all, representation functions are not invertible, so we cannot recover the original example from its intermediate representation or interpolate between data points. Moreover, the domain of representation functions is the entire data domain $\mathbb{R}^n$. This domain is mostly composed of meaningless noise and data occupy only a thin region inside of it. So, even if representation functions provide us a distance, those metrics are incapable of distinguishing between meaningful data and noise.

We find out that a ReLU neural network implicitly identifies a low-dimensional submanifold of the data domain that contains real data. We prove that if the activation function is piecewise-linear (e.g. ReLU), the neural network decomposes the data domain $\mathbb{R}^n$ as the disjoint union of submanifolds (the leaves of a foliation, using the terminology of differential geometry). The dimension of every submanifold (every leaf of the foliation) is bounded by the number of classes of our classification model, so it is much smaller than $n$, the dimension of the data domain $\mathbb{R}^n$. Our main theoretical result, Theorem 3.1, stems from the study of the properties of a variant of the Fisher Information matrix, the *local data matrix*. However, Theorem 3.1 cannot tell us which leaves of this foliation are meaningful, i.e. what are the possible interesting practical applications of the submanifolds that compose the foliation. The interpretation of this geometric structure can only come from experiments. We report experiments performed on MNIST dataset. We choose to focus on MNIST because it is easily interpretable and because small networks are sufficient to reach a high accuracy.

Our experiments suggest that all valid data points lie on only one leaf of the foliation, the *data leaf*. To observe this phenomenon we take an example from the dataset and we try to connect it with another random example following a path along the leaf containing the starting point. If such a path exists, it means that the destination example belongs to the same leaf of the foliation. Visualizing the intermediate points on these joining paths, we see that the low-dimensional data manifold defined by the model is not the anthropocentric data manifold composed of data meaningful for a human observer. The model-centric data manifold comprises images that do not belong to a precise class. The model needs those transition points to connect points with different labels. At the same time, it understands that such transition points represent an ambiguous digit: on such points, the model assigns a low probability to every class.

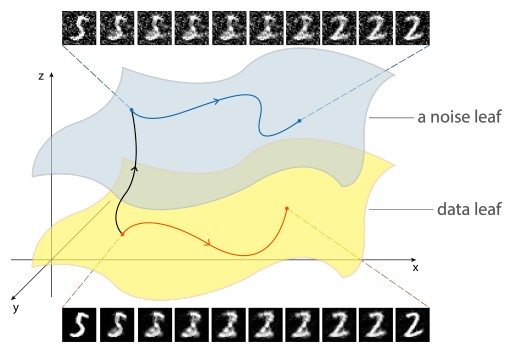

Figure 1: Simplified summary of our experiments: images from MNIST are connected by paths on the *data leaf*, while images on other leaves are noisy.

The experiments also show that moving orthogonally to the data leaf we find noisy images. That means that the other leaves of the foliation contain images with a level of noise that increases with the distance from the data leaf. These noisy images become soon meaningless to the human eye, while the model still classifies them with high confidence. This fact is a consequence of the property of the local data matrix: equation (8) prescribes that the model output does not change if we move in a direction orthogonal to the tangent space of the leaf on which our data is located.

This remark points us to other possible applications of the model-centric data manifold. We could project a noisy point on the data leaf to perform denoising, or we can use the distance from the data leaf to recognize out-of-distribution examples.

The main contributions of the paper are:
1. the definition of the local data matrix $G(x, w)$ at a point $x$ of the data domain and for a given model $w$, and the study of its properties;
2. the proof that the subspace spanned by the eigenvectors with non-zero eigenvalue of the local data matrix $G(x, w)$ can be interpreted as the tangent space of a Riemannian manifold, whose dimension is bounded by the number of classes on which our model is trained;
3. the identification and visualization of the model-centric data manifold through paths, obtained via experiments on MNIST.

**Organization of the paper**. In Section 2, we review the fundamentals of information geometry using a novel perspective that aims at facilitating the comprehension of the key concepts of the paper. We introduce the *local data matrix* $G(x, w)$ and we summarize its properties in Prop. 2.1. In Section 3, we show that, through the local data matrix, under some mild hypotheses, the data domain foliates as a disjoint union of leaves, which are all Riemannian submanifolds of $\mathbb{R}^n$, with metric given via $G(x, w)$. In Section 4, we provide evidence that all our dataset lies on one leaf of the foliation and that moving along directions orthogonal to the data leaf amounts to adding noise to data.

## 2 INFORMATION GEOMETRY

Here we collect some results pertaining to information geometry (Amari, 1998; Nielsen, 2018), using a novel perspective adapted to our question, namely how to provide a manifold structure to the space containing data.

Let $p(y|x, w)$ be a discrete probability distribution on $C$ classification labels, i.e. $p(y|x, w) = (p_i(y|x, w))_{i=1,...,C}$, $x \in \Sigma \subset \mathbb{R}^n$, $w \in \mathbb{R}^d$. In the applications, $x$ represent input data belonging to a certain dataset $\Sigma$, while $w$ are the learning parameters, i.e. the parameters of the empirical model. As we are going to see in our discussion later on, it is fruitful to treat the two sets of variables $x$ and $w$ on equal grounds. This will naturally lead to a geometric structure on a low dimensional submanifold of $\mathbb{R}^n$, that we can navigate through paths joining points in the dataset $\Sigma$ (see Section 4).

In order to give some context to our treatment, we define, following Amari (1998) Section 3, the *information loss* $I(x, w) = -\log(p(y|x, w))$ and the *loss function* $L(x, w) = \mathbb{E}_{y \sim q}[I(x, w)]$. Typically $L(x, w)$ is used for practical optimizations, where we need to compare the model output distribution $p(y|x, w)$ with a certain known true distribution $q(y|x)$. We may also view $L(x, w)$ as the Kullback-Leibler divergence up to the constant $-\sum_i q_i(y|x) \log q_i(y|x)$, irrelevant for any optimization problem:

$$L(x, w) = \mathbb{E}_{y \sim q}[-\log(p(y|x, w))] = \sum_{i=1}^{C} q_i(y|x) \log \frac{q_i(y|x)}{p_i(y|x, w)} - \sum_{i=1}^{C} q_i(y|x) \log q_i(y|x) =$$
$$= D_{\mathrm{KL}}(q(y|x)||p(y|x, w)) - \sum_{i=1}^{C} q_i(y|x) \log q_i(y|x) \tag{1}$$

A popular choice for $p(y|x, w)$ in deep learning classification algorithms is

$$p_i(y|x, w) = \mathrm{softmax}(s_1(x, w), \ldots, s_C(x, w))_i = \frac{e^{s_i(x, w)}}{\sum_{j=1}^{C} e^{s_j(x, w)}}, \tag{2}$$

where $s(x, w) \in \mathbb{R}^C$ is a score function determined by parameters $w$. From such $p(y|x, w)$ we derive the cross-entropy with softmax loss function:

$$L(x, w) = \mathbb{E}_{y \sim q}[I(x, w)] = \mathbb{E}_{y \sim q}[-\log p(y|x, w)] = -s_{y_x}(x, w) + \log \sum_{j=1}^{C} e^{s_j(x, w)}, \tag{3}$$

where $L(x, w)$ is computed with respect to the probability mass distribution $q(y|x)$ assigning 1 to the correct label $y_x$ of our datum $x$ and zero otherwise. Other approaches rely on label smoothing (Szegedy et al., 2016), hence they take a different $L(x, w)$. However, since our treatment mainly relies on the expression of $I(x, w)$, our results will also apply to such loss functions, provided some hypotheses, that we list in Section 3, are satisfied.

Going back to the general setting, notice that: $\mathbb{E}_{y \sim p}[\nabla_w I(x, w)] = 0$. In fact,

$$\mathbb{E}_{y \sim p}[\nabla_w(\log(p(y|x, w)))] = \sum_{i=1}^{C} p_i(y|x, w)\nabla_w \log p_i(y|x, w) = \sum_{i=1}^{C} \nabla_w p_i(y|x, w) = \nabla_w 1 = 0. \tag{4}$$

Let us now define the following two matrices:

$$F(x, w) = \mathbb{E}_{y \sim p}[\nabla_w(\log(p(y|x, w))) \cdot \nabla_w(\log(p(y|x, w)))^T] \tag{5}$$

$$G(x, w) = \mathbb{E}_{y \sim p}[\nabla_x(\log(p(y|x, w))) \cdot \nabla_x(\log(p(y|x, w)))^T] \tag{6}$$

We call $F(x, w)$ the *local Fisher matrix* at the datum $x$ and $G(x, w)$ the *local data matrix* given the model $w$. The Fisher matrix (Amari, 1998) is obtained as $F(w) = \mathbb{E}_{x \sim \Sigma}[F(x, w)]$ and it gives information on the metric structure of the space of parameters. Similarly, we can reverse our perspective and see how $G(x, w)$, allows us to recognize some structure in our dataset.

The following observations apply to both $F(x, w)$ and $G(x, w)$ and provide the theoretical cornerstone of Section 3.

**Proposition 2.1.** *Let the notation be as above. Then:*

1. $\ker F(x,w) = \text{span}\{\nabla_w \log p_i(y|x,w)|1 \leq i \leq C\}^{\perp}$;
   $\ker G(x,w) = \text{span}\{\nabla_x \log p_i(y|x,w)|1 \leq i \leq C\}^{\perp}$.

2. $\text{rank}\, F(x,w) < C, \quad \text{rank}\, G(x,w) < C$.

*Proof.* See Appendix B. □

This result tells us that the rank of both $F(x,w)$ and $G(x,w)$ is bounded by $C$, the number of classes in our classification problem. The consequence of the bound on $\text{rank}\, F(x,w)$ is the following: in SGD dynamics with a single example, the number of directions in which the change in our parameters modifies our loss is severely limited. The consequence on the bound on $\text{rank}\, G(x,w)$ is even more striking: it will allow us to define a submanifold of $\mathbb{R}^n$ of dimension $\text{rank}\, G(x,w)$ (Section 3), that our experiments show contains our dataset (Section 4). In practical situations, this dimension is much lower than the size of $G(x,w)$, i.e. the input size $n$, as shown in Table 1.

Table 1: Bound on the rank of $G(x,w)$ for popular image classification tasks.

| **Dataset** | **Size of** $G(x,w)$ | **Bound on** $\text{rank}\, G(x,w)$ |
|---|---|---|
| MNIST | 784 | 10 |
| CIFAR-10 | 3072 | 10 |
| CIFAR-100 | 3072 | 100 |
| ImageNet | 150528 | 1000 |

## 3 THE MODEL VIEW ON THE DATA MANIFOLD

We now turn to examine some properties of the matrices $F(x,w)$ and $G(x,w)$ that will enable us to discover a submanifold structure on the portion of $\mathbb{R}^n$ occupied by our dataset (see Section 4) and to prove the claims at the end of the above section.

We recall that, given a perturbation of the weights $w$, the Kullback-Leibler divergence, gives, to second-order approximation, the following formula:

$$D_{\text{KL}}(p(y|x,w+\delta w)||p(y|x,w)) = (\delta w)^T F(x,w)(\delta w) + \mathcal{O}(||\delta w||^3) \tag{7}$$

Equation (7), together with Proposition 2.1, effectively expresses the fact that during SGD dynamics with a mini-batch of size 1, we have only a very limited number of directions, namely $C-1$, in which the change $\delta w$ affects the loss.

Taking the expectation with respect to $x \sim \Sigma$ on both sides of the equation, we obtain the analogous property for the Fisher matrix $F(w) = \mathbb{E}_{x \sim \Sigma}[F(x,w)]$. While $F(x,w)$ has a low rank, the rank of $F(x)$ is bounded by $|\Sigma|(C-1)$, a number that is often higher than the size of $F(x)$. Thus $F(w)$, when non-degenerate, is an effective metric on the parameter space. It allows us to measure, according to a certain step $\delta w$, when we reach a stable predicted probability and thus the end of model training (see Martens (2020) and refs. within).

It must be however noted that $F(w)$ retains its information content only *away* from the trained model, that is, well before the end of the training phase (see Achille et al. (2018) for an empirical validation of such statements). We are going to see, with our experiments in Section 4, that a similar phenomenon occurs for $G(x,w)$. Our Fig. 2 elucidates the rank behaviour of $G(x,w)$ during training.

We now turn to the local data matrix $G(x,w)$, thus interpreting equation (7) in the data domain $\mathbb{R}^n$. For a perturbation $\delta x$ of the data $x$, we have, up to second order approximation:

$$D_{\text{KL}}(p(y|x+\delta x,w)||p(y|x,w)) = (\delta x)^T G(x,w)(\delta x) + \mathcal{O}(||\delta x||^3) \tag{8}$$

This equation suggests to view $G(x, w)$ as a metric on the data domain. However, because of its low rank (see Proposition 2.1), we need to restrict our attention to the subspace $\ker(G(x, w))^\perp$, where $G(x, w)$ is non-degenerate. $G(x, w)$ allows us to define a *distribution* $\mathcal{D}$ on $\mathbb{R}^n$. In general, in differential geometry, we call distribution on $\mathbb{R}^n$ an assignment:

$$x \mapsto \mathcal{D}_x \subset \mathbb{R}^n, \qquad \forall x \in \mathbb{R}^n$$

where $\mathcal{D}_x$ is a vector subspace of $\mathbb{R}^n$ of a fixed dimension $k$ (see Appendix A for more details on this notion in the general context).

Assume now $G(x, w)$ has constant rank; as we shall see in our experiments, this is the case for a non fully trained model (see Section 4 for more details). We thus obtain a *distribution* $\mathcal{D}$:

$$x \mapsto \mathcal{D}_x = \ker(G(x, w))^\perp \subset \mathbb{R}^n \tag{9}$$

Equation (8) tells us that, if we move along the directions of $\ker(G(x, w))$, the probability distribution $p(y|x, w)$ is constant (up to a second order approximation) while our data is changing. Those are the vast majority of the directions, since $\mathrm{rank}\, G(x, w) < C$ and typically $C << n$, hence, we interpret them as the *noise directions*. On the other hand, if we move from a data point $x$ along the directions in $\ker(G(x, w))^\perp$, data will change along with the probability distribution $p(y|x, w)$ associated with it. These are the directions in which the model moves with confidence; we are going to clarify this key point in Section 4.

We now would like to see if our distribution (9) defines a *foliation structure*. This means that we can decompose $\mathbb{R}^n$ as the disjoint union of submanifolds, called *leaves* of the foliation, and there is a *unique* submanifold (leaf) going through each point $x$ (see Fig. 7 in Appendix A, where the distribution is generated by the vector field $X$ and $\mathbb{R}^2$ is the disjoint union of circles, the leaves of the foliation). The distribution comes into the play, because it gives the tangent space to the leaf through $x$: $\mathcal{D}_x = \ker(G(x, w))^\perp$. In this way, moving along the directions in $\mathcal{D}_x$ at each point $x$, will produce a path lying in one of the submanifolds (leaves) of the foliation.

The existence of a foliation, whose leaf through a point $x$ has tangent space $\mathcal{D}_x$, comes through Frobenius theorem, which we state in the Appendix A and we recall here in the version that we need.

**Frobenius Theorem**. *Let $x \in \mathbb{R}^n$ and let $\mathcal{D}$ be a distribution in $\mathbb{R}^n$. Assume that in a neighbourhood $U$ of $x$:*

$$[X, Y] \in \mathcal{D}, \qquad \forall\, X, Y \in \mathcal{D}. \tag{10}$$

*Then, there exists a (local) submanifold $N \subset \mathbb{R}^n$, $x \in N$, such that $T_z N = \mathcal{D}_z$, for all $z \in N$.*

It is not reasonable to expect that a general classifier satisfies the involutive property (10), however it is remarkable that for a large class of classifiers, namely deep ReLU neural networks, this is the case, with $p$ given by softmax as in equation (2).

**Theorem 3.1.** *Let $w$ be the weights of a deep ReLU neural network classifier, $p$ given by softmax, $G(x, w)$ the local data matrix. Assume $G(x, w)$ has constant rank. Then, there exists a local submanifold $N \subset \mathbb{R}^n$, $x \in N$, such that its tangent space at $z$, $T_z N = \ker(G(z, w))^\perp$ for all $z \in N$.*

*Proof.* See Appendix B. $\qquad\qquad\qquad\qquad\qquad\qquad\qquad\qquad\qquad\qquad\qquad\qquad\qquad\qquad\qquad$ $\square$

Through the application of Frobenius theorem, Theorem 3.1 gives us a foliation of the data domain $\mathbb{R}^n$. $\mathbb{R}^n$ decomposes into the disjoint union of $C - 1$ dimensional submanifolds, whose tangent space at a fixed $x \in \mathbb{R}^n$ is $\mathcal{D}_x = \ker(G(x, w))^\perp$ (see Appendix A). Every point $x$ determines a unique submanifold corresponding to the leaf of the foliation through $x$. We may extend this local submanifold structure to obtain a global structure of manifold on a leaf, still retaining the above property regarding the distribution.

As we shall see in Section 4, we can move from a point $x$ in our dataset $\Sigma$ to another point $x'$ also in $\Sigma$ with an *horizontal* path, that is a path tangent to $\mathcal{D}_x$, hence lying on the leaf of $x$ and $x'$. Our experiments show that we can connect every pair of points $(x, x') \in \Sigma \times \Sigma$ with horizontal paths. It means that all the dataset belongs to a single leaf, which we call the *data leaf* $\mathcal{L}$. Our model, through the local data matrix $G(x, w)$, enables us to move on the low-dimensional submanifold $\mathcal{L}$, to which all of our dataset belongs. Of course not all the points of $\mathcal{L}$ correspond to elements of the dataset;

however as we show in the experiments, on most points of $\mathcal{L}$ the model gives prediction compatible with human observers.

We also notice that each leaf of our foliation comes naturally equipped with a metric given at each point by the matrix $G(x, w)$, restricted to the subspace $\ker(G(x, w))^{\perp}$, which coincides with the tangent space to the leaf, where $G(x, w)$ is non-degenerate. Hence $G(x, w)$ will provide each leaf with a natural Riemannian manifold structure. We end this section with an observation, comparing our approach to the geometry of the data domain, with the parameter space.

**Remark**. Equation (7) provides a metric to the parameter space $\mathbb{R}^d$, motivating our approach to the data domain. For each $w \in \mathbb{R}^d$, we can define, as we did for the data domain, a distribution $w \mapsto \mathcal{D}'_w := \ker(F(w))^{\perp}$, using the Fisher matrix. However, it is easy to see empirically that this distribution is *not involutive*, i.e. there is no foliation and no submanifold corresponding to it.

## 4 EXPERIMENTS

We performed experiments on the MNIST dataset (LeCun et al., 1998) and on the CIFAR-10 dataset (Krizhevsky et al., 2010). We report in this section the experiments on the MNIST dataset only, because they are easier to interpret in the geometrical framework (foliation and leaves) introduced in our previous section. CIFAR-10 experiments are reported in the dedicated Appendix C. Thus, all the following experiments use a simple CNN classifier trained on the MNIST.

The definition of distribution in Section 3 and the consequent results require the rank of $G(x, w)$ to be constant on every point $x$ for a certain parameter configuration $w$. We remark that the rank of $G(x, w)$ can be at most $C - 1$ (Proposition 2), i.e. much lower than the size of $G(x, w)$ (Table 1), so we can expect that $\operatorname{rank} G(x, w) = C - 1$ for almost all $x \in \mathbb{R}^n$.

We observe that during the training of the CNN, the rank of $G(x, w)$ is stable and equal to $C - 1$. That changes at the end of the training, when the model reaches convergence. At that point, the output class probability distribution on most images is highly unbalanced and sparse and all the eigenvalues of $G(x, w)$ tend to 0. Figure 2 shows the decreasing trend of the mean trace of $G(x, w)$ on a batch during the training. The plot is similar to the ones reported in Achille et al. (2018) for the Fisher information matrix and it tells us that the local data matrix $G(x, w)$ loses its informative content at the end of training as well. This observation motivates us to perform subsequent experiments on a partially trained model and with data points where every output class probability is lower than 0.99.

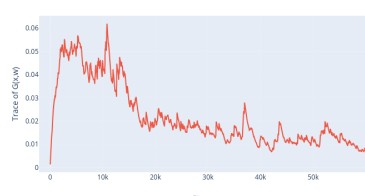

Figure 2: Mean trace of $G(x, w)$ during training

Since a complex model reaches convergence on MNIST digit classification too fast, we have resorted to a very small CNN. The network used in the following experiments has 18,982 parameters and ReLU activation function. It is similar to LeNet, but with 4 channels in both the two convolutional layers and 32 hidden units in the fully-connected layer. We train this network with SGD for just one epoch to prevent model convergence.

Such a neural network classifier satisfies all the hypothesis of Theorem 3.1, so it views the data domain $\mathbb{R}^n$ as a *foliation*. However, this result does not give us any clue about the characterization of leaves. We know, however, that, moving away from a data point $x$ while remaining on the same leaf, we can obtain points with different labels, while moving in a direction orthogonal to the tangent space to the leaf of $x$, we obtain points with the same label as $x$, as long as the estimate (8) holds.

To understand the distinguishing factors of a leaf, we move from a data point $x$ across leaves and we inspect the evolution. We start from an image in MNIST test set and we let it evolve moving orthogonally to leaves. In order to get further and further away, we use a fixed random direction and at every step we project it on the space orthogonal to the current leaf. From the previous section, we know that such a space coincides with $\ker G(x, w)$.

We observe in Figure 3, that the noise in the images increases steadily. Moreover, we can see that, as predicted by equation (8), model predictions remain very certain even when the digit is indistinguishable for the human eye.

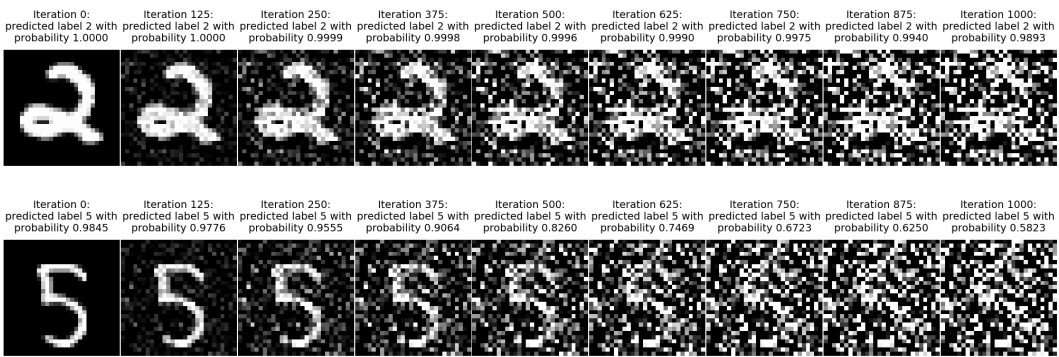

Figure 3: Paths across leaves of the foliation from a valid image.

This experiment makes us speculate that a leaf is characterized by a constant amount of noise. If that is the case, all valid data should reside on the same leaf characterized by the absence of noise, the data leaf. To verify our intuition we follow *horizontal* paths connecting two images from MNIST test set. If it is possible to join two inputs with a horizontal path, then we know that those points are on the same leaf. In our case a horizontal path is tangent to the distribution $\mathcal{D}$ described in the previous section, i.e. $\mathcal{D}_x = (\ker G(x,w))^\perp = \text{span}\{\nabla_x \log p_i(y|x,w) | 1 \le i \le C\}$ from Proposition 1. We remark that $\mathcal{D}_x$ coincides with the row space of the Jacobian matrix $\text{Jac}_x \log p(x,w)$. We will use this equivalence to efficiently compute the projection of a vector on the distribution $\mathcal{D}_x$.

The algorithm to approximate a path from a source point to a destination point is a simplification of Riemannian gradient descent (Gabay, 1982) using the euclidean distance from the destination point as loss function. Since we do not have an explicit characterization of the leaf, we cannot perform the retraction step commonly used in optimization algorithms on manifold. To circumvent this problem, we normalize the gradient vector and use a small step size of 0.1 in our experiments. The normalization assures that the norm of the displacement at every step is controlled by the step size. The iterative algorithm is reported in Algorithm 1.

---

**Algorithm 1:** Find a horizontal path between source and destination points.

**Input:** $s$: source; $d$: destination; $\alpha$: step size, $T$: number of iterations; $w$: model parameters.

1 $x_0 \leftarrow s$;
2 **for** $t = 1, \ldots, T$ **do**
3     $j \leftarrow \text{Jac}_x \log p(x_{t-1}, w)$ (Calculate Jacobian of $\log p(x_{t-1}, w)$ w.r.t. $x_{t-1}$);
4     $v \leftarrow \text{projection}(d - x_{t-1}, j)$ (Project the gradient of $||d - x_{t-1}||^2$ on $\mathcal{D}_{x_{t-1}}$);
5     $x_t \leftarrow x_{t-1} + \alpha \frac{v}{||v||}$ (Update $x_{t-1}$ to obtain the next point $x_t$ on the horizontal path);
6 **return** $\{x_t\}_{t=0,\ldots,T}$

---

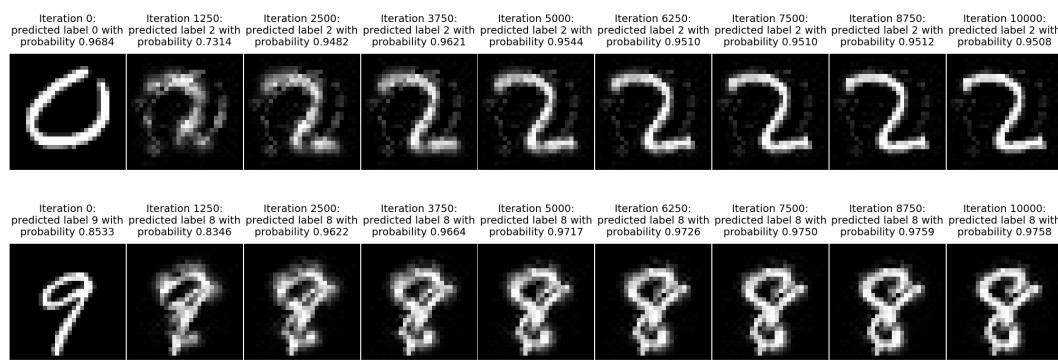

Figure 4: Horizontal paths between two images in MNIST test set. Here and in all the following images the pairs of source-destination points are sampled randomly from the test set.

Figure 4 shows the results of the application of Algorithm 1 on some pairs of images in MNIST test set. We observe that it is possible to link very different images with an horizontal path, confirming our conjecture about the existence of the data leaf. This experiment shows that the model sees the data on the same manifold, namely one leaf of the foliation determined by our distribution $\mathcal{D}$.

The observation of the paths on the data leaf gives us a novel point of view on the generalization property of the model. First of all, while we could expect to find all training data on the same leaf, it is remarkable that test data are placed on the same leaf too. Furthermore, we observe that the data leaf is not limited to digits and transition points between them. In fact, we can find paths connecting valid images with images of non-existent digits similar to letters, as shown in Figure 5. That fact suggests that the model-centric data manifold is somewhat more general than the classification task on which the model is trained.

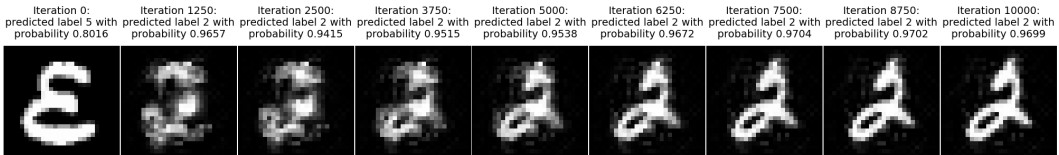

Figure 5: Horizontal path between a mirrored image and a valid image from MNIST test set.

Our final experiment gives us another remarkable confirmation of our theoretical findings: it is impossible to link with a horizontal path a noisy image outside the data leaf and a valid data image. Figure 6 shows that even after thousands of iterations of Algorithm 1, it is impossible to converge to the destination point. Indeed, the noise is preserved along the path and the noise pattern stabilizes after 5000 iterations. The final point reached is a noisy version of the actual destination point.

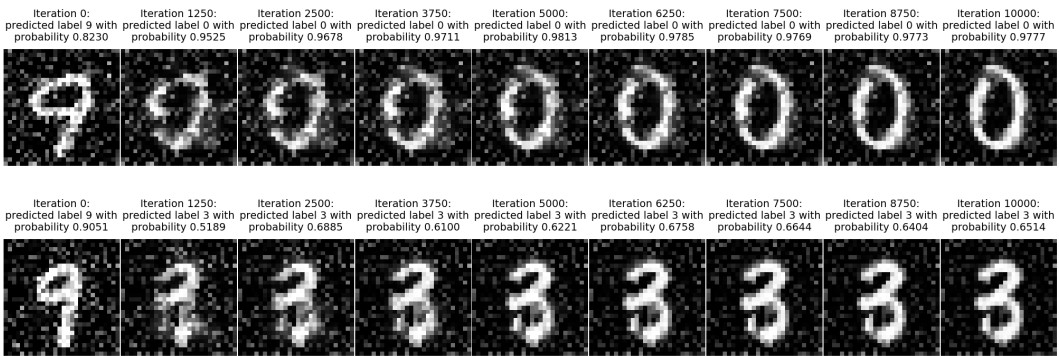

Figure 6: Horizontal paths unable to reach a valid image in MNIST test set from a noisy image.

All those experiments confirm that there exists a model-centric data manifold, the data leaf of the foliation. In addition they exhibit that the leaves in the foliation of the data domain are characterized by the noise content.

## 5 RELATED WORKS

**Fisher Information Matrix**. In Achille et al. (2018) the authors discuss the information content of the Fisher matrix, and they show that such content changes during the training of a neural network, decreasing rapidly towards the end of it. This shows that the Fisher-Rao metric acquires importance during the training phase only (see also Kirkpatrick et al. (2017)). More on the geometry of the parameter space, as a Riemannian manifold, is found in Sommer & Bronstein (2020). In this paper, we take an analog of the Fisher-Rao metric, but on the data domain. In our examples, we see a phenomenon similar to the one observed in Achille et al. (2018): the trace of the local data matrix decreases rapidly, as the model completes its training. Other metrics on datasets were suggested, for example see Dukler et al. (2019) and refs. within, but with different purposes. Here, our philosophy

is the same as in Bergomi et al. (2019): we believe that data itself is not equipped with a geometric structure, but such structure emerges only when the model views data, with a given classification task.

**Intrinsic Dimension**. Ansuini et al. (2019) measure the intrinsic dimension of layer representations for many common neural network architecture. At the same time, they measure the intrinsic dimension of MNIST, CIFAR-10 (Krizhevsky et al., 2010) and ImageNet (Deng et al., 2009) datasets. Our objective is similar, but we do not specifically quantify the dimension of the data manifold. The data leaf reflects how a classifier sees a geometric structure on the discrete data points from a dataset. Its dimension is intimately linked to the classification task. For this reason the results are not directly comparable.

Ansuini et al. (2019) show that MNIST and neural networks trained on it behave very differently from networks trained on CIFAR-10 or ImageNet. While our experiments focus on MNIST, additional experiments on CIFAR-10 are shown in the Appendix C.

**Adversarial Attacks**. Our method to navigate the leaves of the foliation is very similar to common adversarial attack methods, like Fast Gradient Sign Method (Goodfellow et al., 2015) or Projected Gradient Descent. Adversarial attacks and our navigation algorithm both rely on gradients $\nabla_x \log p_i$, but adversarial generation algorithms perturb the original image by $\mathrm{sign}(\nabla_x \log p_i)$. In general, $\mathrm{sign}(\nabla_x \log p_i) \notin \ker(G(x, w))^{\perp}$, so adversarial examples are created perturbing the image outside the data leaf.

## 6  CONCLUSIONS

In this paper, we introduce the local data matrix, a novel mathematical object that sheds light on the internal working of a neural network classifier. We prove that the model organizes the data domain according to the geometric structure of a foliation. Experiments show that valid data are placed on the same leaf of the foliation, thus the model sees the data on a low-dimensional submanifold of the data domain. Such submanifold appears more general than the model itself, because it includes meaningless, but visually similar, images together with training and test data.

In the future, we aim to characterize the data leaf and to study the Riemannian metric given by the local data matrix. If we could analytically characterize the leaves of the foliation by the degree of noise, we could distinguish noisy data from valid examples. That can be used in the inference phase to exclude examples on which model predictions are not reliable. Furthermore, it could pave the way to the creation of novel generic denoising algorithms applicable to every kind of data.

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

## A    APPENDIX: FROBENIUS THEOREM

For the reader's convenience, we collect here few facts regarding differentiable manifolds, for more details see Abraham et al. (2012), Tu (2008).

Let $M$ be a differentiable manifold. Its tangent space $T_xM$ at a point $x \in M$ can be effectively defined as the vector space of derivations at $x$, so $T_xM = \{\sum_i a^i \partial_i\}$, after we choose a chart around $x$. A *vector field* $X$ on $M$ is a function $x \mapsto X_x \in T_xM$ assigning to any point of $x \in M$ a tangent vector $X_x$. An *integral curve* of the vector field $X$ at $p$ is a map $c : (-\epsilon, \epsilon) \longrightarrow M$, such that $c(0) = p$ and $\frac{d}{dt}c(t) = X_{c(t)}$, for all $t \in (-\epsilon, \epsilon), \epsilon \in \mathbb{R}$.

For example, in $\mathbb{R}^2$, we can define the vector field:

$$X : (x, y) \mapsto -y\partial_x + x\partial_y$$

One may also view $X$ as assigning to a point $p = (x, y)$ in $\mathbb{R}^2$ the vector $(-y, x)$ applied at $p$. The integral curve of $X$ at $(1, 0)$ is the circle $c(t) = (cos(t), sin(t))$, $t \in (-\pi, \pi)$, in fact $c'(t) = (-sin(t), cos(t)) = X_{c(t)}$. More in general, the integral curve of $X$ at a generic point $p$ is the circle centered at the origin and passing through $p$. If $TM = \coprod_{x \in M} T_xM$ (disjoint union) denotes the

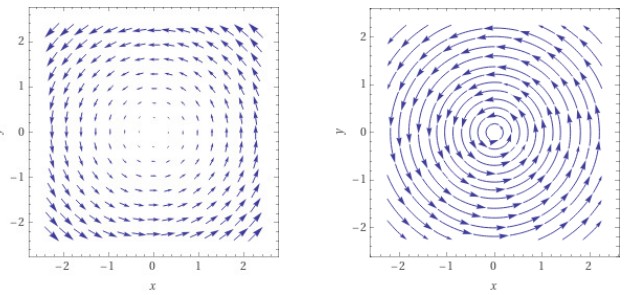

Figure 7: Vector field $X = -y\partial_x + x\partial_y$ in $\mathbb{R}^2$ and its integral curves.

*tangent bundle*, we also call the vector field $X$ a *section* of $TM$.

Since $T_xM$ is the vector space of derivations on differentiable functions defined on a neighbourhood of $x$, we can write $X_x(f)$; this is the result of the application of the derivation $X_x \in T_xM$ to the function $f \in C_M^\infty(U)$, where $C_M^\infty(U)$ denotes the differentiable functions $f : U \longrightarrow \mathbb{R}$, $x \in U$ open in $M$. As $x$ varies in $U$, $X_x(f)$ defines a function, which we denote with $X(f)$.

We call $\chi(M)$ the *smooth* vector fields on $M$, that is, those vector fields expressed in local coordinates as $X_x = \sum_i a_i(x)\partial_i$, with $a_i : U \longrightarrow \mathbb{R}$ smooth functions, $x \in U$, $U$ open in $M$.

We can define a *bracket* on the space of vector fields as follows:

$$[X, Y](f) := X(Y(f)) - Y(X(f)), \qquad f \in C_M^\infty(M), \quad X, Y \in \chi(M)$$

$[\,,\,]$ is bilinear and satisfies:
1. $[X, Y] = -[Y, X]$ (antisymmetry);
2. $[X, [Y, Z]] + [Y, [Z, X]] + [Z, [X, Y]] = 0$ (Jacoby identity).

We are ready to define *distributions*, which play a key role into our treatment.
Let $M$ be a differentiable manifold. We define a *distribution* $\mathcal{D}$ of rank $k$ an assignment:

$$x \longrightarrow \mathcal{D}_x \subset T_xM, \qquad \forall x \in M$$

where $\mathcal{D}_x$ is a subspace of $T_xM$ of dimension $k$. We say that $\mathcal{D}$ is *smooth* if at each point $x \in M$, there exists an open set $U$, $x \in U$, $\mathcal{D}_y$ is spanned by smooth vector fields on $U$ for all $y \in U$.

We say that a distribution is *integrable* if for any $x \in M$ there exists a local submanifold $N \subset U$, $U$ open in $M$, $x \in N$, called a *local integrable manifold* such that $T_zN = \mathcal{D}_z$ for all $z \in N$. When such $N$ exists globally, we say we have a *foliation* of $M$, that is we can write $M$ as the disjoint union of integrable submanifolds, all of the same dimension and immersed in $M$. Each integrable submanifold of the foliation is called a *leaf*. For example in $\mathbb{R}^3 \setminus \{(0, 0, 0)\}$, the distribution:

$$p = (x, y, z) \mapsto \mathcal{D}_p = (\text{span}\{x\partial_x + y\partial_y + z\partial_z\})^\perp \tag{11}$$

is integrable: at each point $p$, $\mathcal{D}_p$ is the tangent plane to a sphere centered at the origin and passing through $p$. Hence we can write the space $\mathbb{R}^3 \setminus \{(0,0,0)\}$ as the disjoint union of spheres: each sphere is a leaf of the foliation thus obtained. We may also say that $\mathbb{R}^3 \setminus \{(0,0,0)\}$ *foliates* as the disjoint union of spheres. Another example is given in Figure 7, showing how the distribution generated by the vector field $X$ foliates $\mathbb{R}^2 \setminus \{(0,0)\}$ as the disjoint union of circles.

We say that a distribution is *involutive* if for all vector fields $X, Y \in \mathcal{D}$ we have $[X, Y] \in \mathcal{D}$.

Frobenius Theorem establishes an equivalence between these two key properties of a distribution, namely integrability and involutivity, thus giving us an effective method to establish when a distribution gives a foliation of $M$ into disjoint submanifolds.

**Theorem A.1. Frobenius Theorem** *Let $M$ be a differentiable manifold and $\mathcal{D}$ a distribution on $M$. Then $\mathcal{D}$ is involutive if and only if it is integrable. If this occurs, then there exists a foliation on $M$, whose leaves are given by the integrable submanifolds of $\mathcal{D}$.*

*Proof.* See Abraham et al. (2012) Section 4.4. □

Frobenius Theorem tells us that an involutive distribution on $M$ defines at each point $p$ a submanifold $N$ of dimension $k$ of $M$. This means that, locally, we can choose coordinates $(x_1, \ldots, x_n)$ for $M$, so that, in the chart neighbourhood of $p$, the submanifold $N$ is determined by equations $x_{k+1} = c_{k+1}, \ldots, x_n = c_n$, where $c_i$ are constants. We have then that $(x_1, \ldots, x_k)$ are local coordinates for the submanifold $N$ around $p$. Notice that the vector fields associated to these particular coordinates $X_i = \partial_{x_i}$ verify a stronger condition than involutivity, namely $[X_i, X_j] = 0$. Furthermore, the proof of Frobenius theorem is constructive: it will give us the vector fields $X_i$ and the coordinates $x_i$ as their integral curves. Let us see in a significant example, what this construction amounts to.

We express the distribution $\mathcal{D}$ defined in (11) explicitly as:

$$
\begin{aligned}
p = (x, y, z) \mapsto \mathcal{D}_p &= (\text{span}\{x\partial_x + y\partial_y + z\partial_z\})^\perp = \\
&= \text{span}\{X = y\partial_x - x\partial_y, Y = z\partial_x - x\partial_z\}
\end{aligned}
\tag{12}
$$

As one can check $[X, Y]_p = -y\partial_z + z\partial_y \in \mathcal{D}_p$ for all $p \in \mathbb{R}^3 \setminus \{(0,0,0)\}$. With Gauss reduction we transform

$$
\begin{pmatrix} y & -x & 0 \\ z & 0 & -x \end{pmatrix} \longrightarrow \begin{pmatrix} 1 & 0 & -x/z \\ 0 & 1 & -y/z \end{pmatrix}
$$

One can then verify that $[X_1, X_2] = 0$ for $X_1 = \partial_x - (x/z)\partial_z$, $X_2 = \partial_y - (y/z)\partial_z$.

The calculations of this example are indeed a prototype for the treatment in the general setting: given a local basis for an involutive $\mathcal{D}$, that is a set of vector fields $X_1, \ldots X_k$, which are linearly independent in a chart neighbourhood, using Gauss algorithm, we can transform them into another basis $Y_1 \ldots Y_k$ with the property $[Y_i, Y_j] = 0$. The integral curves of the $Y_i$'s will then give us the coordinates for the leaf manifold in the given chart.

## B  APPENDIX: PROOFS

In this appendix we collect the proof of the mathematical results stated in our paper. Recall the two key definitions of local Fisher and data matrices:

$$
F(x, w) = \mathbb{E}_{y \sim p}[\nabla_w(\log(p(y|x,w))) \cdot \nabla_w(\log(p(y|x,w))^T]
\tag{13}
$$

$$
G(x, w) = \mathbb{E}_{y \sim p}[\nabla_x(\log(p(y|x,w))) \cdot \nabla_x(\log(p(y|x,w))^T]
\tag{14}
$$

We start with the statement and proof of Proposition 2.1.

**Proposition B.1.** *Let the notation be as in Section 2. Then:*

1. $\ker F(x, w) = \text{span}\{\nabla_w \log p_i(y|x,w)|1 \leq i \leq C\}^\perp$;
   $\ker G(x, w) = \text{span}\{\nabla_x \log p_i(y|x,w)|1 \leq i \leq C\}^\perp.$

2. $\text{rank } F(x, w) < C, \quad \text{rank } G(x, w) < C.$

*Proof.* We prove the results for $F(x, w)$, the proofs for $G(x, w)$ are akin.

*1.* We show that $\ker F(x, w) \subseteq \text{span}\{\nabla_w \log p_i(y|x, w) | 1 \le i \le C\}^\perp$:

$$u \in \ker F(x, w) \Rightarrow u^T F(x, w) u = 0 \Rightarrow \mathbb{E}_{y \sim p} \left[ \langle \nabla_w \log p(y|x), u \rangle^2 \right] = 0$$
$$\Rightarrow \langle \nabla_w \log p_i(y|x), u \rangle = 0 \quad \forall\, i = 1, \ldots, C. \tag{15}$$

On the other hand, if $u \in \text{span}\{\nabla_w \log p_i(y|x, w) | 1 \le i \le C\}^\perp$ then $u \in \ker F(x, w)$:

$$F(x, w) u = \mathbb{E}_{y \sim p} \left[ \nabla_w \log p_i(y|x) \langle \nabla_w \log p_i(y|x), u \rangle \right] = 0. \tag{16}$$

*2.* From 2.1.1, we know that $\text{rank } F(x, w) \le C$. Equation (4) allows us to conclude the proof. $\square$

We now go to the main mathematical result of our paper, which is the key to provide with a manifold structure the space of data.

**Theorem B.2.** *Let $w$ be the weights of a deep ReLU neural network classifier, $p$ given by softmax, $G(x, w)$ the local data matrix. Assume $G(x, w)$ has constant rank. Then, there exists a local submanifold $N \subset \mathbb{R}^n$, $x \in N$, such that its tangent space at $z$, $T_z N = \ker(G(z, w))^\perp$ for all $z \in N$.*

*Proof.* By Frobenius Theorem, we need to check the involutivity property (10) for the distribution $x \mapsto \mathcal{D}_x = \ker(G(x, w))^\perp$ on $\mathbb{R}^n$. Since by Prop. 2.1,

$$\mathcal{D}_x = \ker(G(x, w))^\perp = \text{span}\{\nabla_x \log p_k(y|x, w) \,|\, 1 \le k \le C\}$$

we only need to show that:

$$[\nabla_x \log p_i(y|x, w), \nabla_x \log p_j(y|x, w)] \in \text{span}\{\nabla_x \log p_k(y|x, w) \,|\, 1 \le k \le C\}$$

By standard computations, we see that:

$$[\nabla_x \log p_i(y|x, w), \nabla_x \log p_j(y|x, w)] = \mathbb{H}(\log p_i(y|x, w)) \nabla_x \log p_j(y|x, w) +$$
$$- \mathbb{H}(\log p_j(y|x, w)) \nabla_x \log p_i(y|x, w), \tag{17}$$

where $\mathbb{H}(f)$ denotes the Hessian of a function $f$. Here we are using the fact that

$$\left[ \sum_i a \partial_i, \sum_j b_j \partial_j \right] = \sum_{i,j} (a_i \partial_i b_j - b_i \partial_i a_j) \partial_j$$

With some calculations, we have:

$$\mathbb{H}(\log p_i(y|x, w)) = \frac{\mathbb{H}(p_i(y|x, w))}{p_i(y|x, w)} - \nabla_x \log p_i(y|x, w) \cdot (\nabla_x \log p_i(y|x, w))^T \tag{18}$$

and

$$\nabla_x p_i(y|x, w) = \sum_{k=1}^C p_i(y|x, w)(\delta_{ik} - p_k(y|x, w)) \nabla_x s_k, \tag{19}$$

where $s$ is the score function and we make use of the fact $p_i(y|x, w)$ is given by softmax. Notice $\partial_{s_k} p_i(y|x, w) = p_i(y|x, w)(\delta_{ik} - p_k(y|x, w))$, where $\delta_{ik}$ is the Kronecker delta. By (18) and (19):

$$\mathbb{H}(p_i(y|x, w)) = \text{Jac}(\nabla_x p_i(y|x, w)) = \sum_{k=1}^C \nabla_x \left[ p_i(y|x, w)(\delta_{ik} - p_k(y|x, w)) \right] (\nabla_x s_k)^T, \tag{20}$$

where we use the fact that $s_k$ is piecewise linear so that its Hessian is zero, where it is defined. Hence:

$$\mathbb{H}(p_i(y|x, w)) = \frac{\nabla_x p_i(y|x, w)(\nabla_x p_i(y|x, w))^T}{p_i(y|x, w)} - p_i \sum_{k=1}^C \nabla_x p_k(y|x, w)(\nabla_x s_k)^T. \tag{21}$$

Now, in view of (17), using (18) and (21) we compute the expression:

$$\mathbb{H}(\log p_i(y|x, w)) = \frac{\nabla_x p_i(y|x, w)}{p_i(y|x, w)} \left( \frac{\nabla_x p_i(y|x, w)}{p_i(y|x, w)} \right)^T - \sum_{k=1}^C \nabla_x p_k(y|x, w)(\nabla_x s_k)^T +$$
$$- \nabla_x \log p_i(y|x, w) \cdot (\nabla_x \log p_i(y|x, w))^T = - \sum_{k=1}^C \nabla_x p_k(y|x, w)(\nabla_x s_k)^T. \tag{22}$$

Hence:

$$\mathbb{H}(\log p_i(y|x, w))\nabla_x \log p_j(y|x, w) = -\sum_{k=1}^{C} \nabla_x p_k(y|x, w)(\nabla_x s_k)^T \nabla_x \log p_j(y|x, w).$$

Since $(\nabla_x s_k)^T \nabla_x \log p_j(y|x, w)$ is a scalar, we obtain $\mathbb{H}(\log p_i(y|x, w))\nabla_x \log p_j(y|x, w)$ is a linear combination of vectors $\nabla_x p_k(y|x, w)$, and therefore a linear combination of $\nabla_x \log p_k(y|x, w)$. The same holds for $[\nabla_x \log p_i(y|x, w), \nabla_x \log p_j(y|x, w)]$ as we wanted to show. □

## C  APPENDIX: CIFAR-10 EXPERIMENTS

We perform some experiments with the CIFAR-10 dataset and a neural network similar to VGG-11.

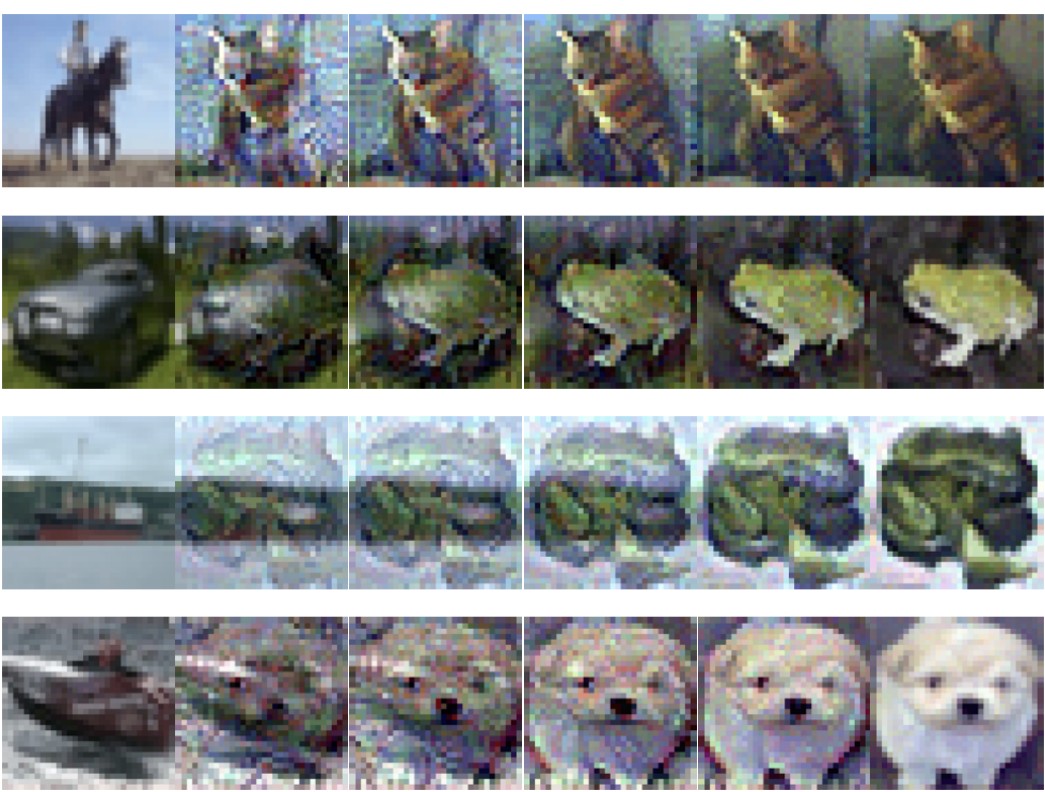

Figure 8: Horizontal paths between images in CIFAR-10 test set.

