# OpenReview forum: " Model-centric data manifold: the data through the eyes of the model"
_ICLR.cc/2021/Conference — Reject_

### Official Review · AnonReviewer4 · 2020-10-27
**Promising contributions, but needs improvements**

**Rating:** 5
**Confidence:** 3

**Review:**

This paper provides an information geometric view on deep ReLU neural network
classifiers which use the softmax function. For a fixed model it defines the
``local data matrix'' based on gradients \( \nabla _x \log p(y|x, w) \) of
conditional log probabilities combining ideas from both scores and the Fisher
information matrix. This matrix gives rise to a Riemannian manifold with a
specific foliation of interest. Experiments using the MNIST dataset suggest
that valid data live in a single leaf.

The finding that involutive property is satisfied for the distribution D_x
connected to the localhost Fisher matrix G is to me quite surprising and an
interesting result. With the constant rank assumption, this indeed implies a
foliation of the data space. It would be nice if the authors could make it
clearer what is the interpretation of the local Fisher matrix, i.e. what it
actually models. This is attempted experimentally but to a lesser degree
theoretically.

Experimental results are presented to validate different hypotheses:
Experiment 1 attempts to validate the assumption of the local data matrix
having constant rank. The authors observe that it breaks for models reaching
convergence. The quantitative relation to Figure 1 is not clear to me and I
did not find the experiment readily reproducible from the source code
provided. The remaining experiments look at paths between pairs of images
from the MNIST test set and images moved along a path. It is not clear how
the 1-2 examples presented for each of the experiments have been selected.
Considering them as ``remarkable confirmation of [the] theoretical findings''
of an impossibility statement feels like an overstatement.

It would be nice if the authors could comment on how an example where the
data is distributed to fill the entire data space can be foliated into C-1
leafs with the data being contained in one of those leafs. This leaf would
then need to fill the entire data space. Is this a case where the constant
rank assumption wails?

The written exposition does not feel ready for presentation yet. Smaller
mistakes (a/an, plural/singular, lie/lay) and notational issues (e.g. missing
nabla in the line before eq. (4)) impede readability. The introduction
ignores generative models like (deep) Boltzmann machines or the Helmholtz
machine and is partially inaccurate (generative models providing an explicit
transformation). The paper's goal or contribution does not become clear from
the abstract.

Overall, in my view, the paper does contain promising contributions, however,
in its current state, it is somewhat premature and the presentation is
(slightly) below the acceptance threshold for ICLR.

---

> ### Author Response · Authors · 2020-11-22
> **Answer to Reviewer 4**
>
> We answer according to the paragraphs of the review. Please see also our collective answer to the common issues raised by all Reviewers.
>
> -- Answer to paragraph 1, 2: We have added in Sec. 2 and 3 some comments to the interpretation of the local Fisher matrix. In particular, in the experiments we do not see the same involutivity property that allows us to apply the Frobenius theorem  and obtain a foliation as it happens for the local data matrix.
>
> -- Answer to paragraph 3: this is a very important issue, we thank the Reviewer for helping us to focus on a key point of our paper that we were unable to convey effectively in our first draft. As we now stress in the introduction and Sec. 3 and 4, the Frobenius Theorem alone will not give us the data leaf, but will only show how R^n foliates into the disjoint union of manifolds. Eq (8) says that if we move away from one leaf, orthogonally, the label of our data will not change, while moving on a leaf we can see that labels change. The experiments then show that, starting from a datum x, we can reach any other datum in our data set, while remaining on the same leaf of the foliation (data leaf). On the other hand, if we try to connect some point in a leaf, which is not the data leaf, to a data point, with a path entirely lying on that leaf, we do not succeed! This is evidence to us that the data leaf contains all data points.
>
> -- Answer to paragraph 4: we believe the data leaf will contain more than just the data point. We have tried to explain this with visual examples, where the paths in the data leaf connecting to images in our data set also contain other images which are clearly not belonging to any class of MNIST (or CIFAR10 as we show in our new Appendix C). We believe that the addition of more figures and explanation has helped us to clarify this important point raised by the Reviewer.
>
> -- Answer to paragraph 5: we have made an effort to better proofread our manuscript and correct our typos. We have also made more precise comments about generative models in our introduction. We find this comment of the Reviewer very inspiring and related to the question: what is the correct metric (analogue of our local Fisher and data matrices) for the dynamics of the parameters of a deep/restricted Boltzmann machine? As the Reviewer suggests this line of investigation appears promising since RBM and BM in general behave very well as generative models and can be tested in small examples like MNIST or similar. We plan to explore this line of research and we thank the Reviewer for this interesting input.

---

### Official Review · AnonReviewer1 · 2020-10-30
**Important theory but "may not be" practical**

**Rating:** 6
**Confidence:** 5

**Review:**

In this work, the authors showed that deep ReLU networks can model the low dimensional manifold structure of the dataset. The authors first define a local data matrix G which is analogous to Fisher matrix. Then they proved that the tangent space of the data manifold is spanned by the eigen vectors of G corresponding to non-zero eigen values. The authors visualize this data manifold on MNIST data.

Below, I present some of the key points (strengths/weaknesses) for this paper. The concrete theoretical result to characterize the data manifold based on the span of the eigen vectors of the data matrix is novel and an important result. On the other hand, simple demonstration on MNIST may not be sufficient to conclude the effectiveness on other natural images, e.g., cifar, imagenet. The authors in [1] argued that for simple dataset like MNIST, the lower dimensional representation may be flat but for natural images, the lower dimensional representation may possibly be curved. This makes one wonder the applicability of such kind of path analysis for natural images. Although, I believe the theoretical contribution is nice, the lack of empirical validation/evaluation weakens the scope of the paper, which justifies the rating ````"6".

[1] https://papers.nips.cc/paper/8843-intrinsic-dimension-of-data-representations-in-deep-neural-networks.pdf

1. It is always a better idea to explain rationale and motivation of a Proposition before stating and proving it, e.g., while reading Proposition 1 for the first time, I have no idea the need for looking at kernel space!

2. The main theorem stating that for deep ReLU network, then the submanifold we get with the tangent space as orthogonal space of the kernel of the data fisher matrix is I believe a very important result.

3. Although I am very impressed with the theoretical result presented in the paper, in practice constancy of rank may not be a valid assumption and as mentioned in section 4 that towards the end the fisher matrix becomes close to a null matrix, the impracticality of the paper comes from the usage of partially trained model. For example, it is not realistic to assume (1) a given network is partially trained (2) not too deep so doesn't converge fast. These in my opinion are bug assumptions and is a bottleneck for the applicability of this paper.

4. Although the justification of not using retraction is meaningful, not sure the need for normalizing gradient, is it to get a ``good grip'' on learning rate?

5. The implication of Fig. 4 showing that the data manifold may include points outside our training/test data is a good observation but I believe things are too simplistic and "too good" for MNIST data. Any comment on dataset with natural images like Cifar10, ImageNet etc.?

6. How the authors compute G(x,w), i.e., Jacobian?

7. In algorithm 1, line 4, where is j used, is it to compute D_x?

---

> ### Author Response · Authors · 2020-11-22
> **Answer to Reviewer 1**
>
> We answer according to the numbering of the review. Please see also our collective answer to the common issues raised by all Reviewers.
>
> -- Answer to (1): indeed we have added more explanation to our mathematical treatment (see collective answer).
>
> -- Answer to (2): we have quoted and highlighted our main result in the introduction, so that it is easier to focus on it.
>
> -- Answer to (3): our assumptions are valid for models on MNIST and on CIFAR10 (we added this model to our treatment, see Appendix C).
>
> -- Answer to (4): We use a typical algorithm and learning rate to train our model.
>
> -- Answer to (5): We have added Appendix C to see what happens for CIFAR10, our results appear confirmed, we can connect two data images by a path on the data leaf, though it is harder to discern noise in this data set.
>
> -- Answer to (6, 7): We have put more details in Sec. 3. In brief, the Jacobian is computed applying the back-propagation algorithm with respect to data. We never need to compute explicitly the full $G(x,w)$, the Jacobian only is sufficient. In particular, the rows of the Jacobian are used to compute a basis of $D_x$.

---

### Official Review · AnonReviewer3 · 2020-11-03
**Official Blind Review #3**

**Rating:** 4
**Confidence:** 4

**Review:**

Update after response:
While I appreciate the authors' attempt to address my concerns, the fact that model is required to not be fully trained is concerning. It was in this context that I suggested label smoothing - that training on smoothed labels might address a sparse G matrix, but it seems like this point was not communicated clear enough by me and not understood by the authors.

The authors suggest some applications of the model centric view of the data, but do not present any experiments regarding these applications. I believe the paper might be more convincing if those experiments are added in the next version of the paper.

At this time however I will still have to vote for rejection.

Original Review:

This paper describes a manifold construction of the space of inputs to a deep ReLU network, by defining a metric using the gradient of the network's loss function with respect to its input. The paper uses the Frobenius Theorem to show that every point in the input space R^n can be associated with a submanifold which constitutes "the model's view of the data."

My main concern with this work is that the import of this work is not clear. At the current stage, the authors have only identified a foliation of R^n (not the data manifold, since that is presumably only a low dimensional manifold of R^n). Do the authors envision using their manifold construction as a generative model? Or will this view of the input space assist in identifying adversarial examples or coming up with defenses? Can this approach help in identifying structures within the data?

While the use of Frobenius Theorem is new, this line of research of visualizing the "manifold" of data "preferred" by the model goes back to the deep dream visualizations from google and other efforts to understand the representations learned by deep networks. The authors do not refer to that line of work, and I would be curious to learn how they see their work differing from visualizations of features. Perturbing images along gradients of the log likelihood is also a common technique (Fast Gradient Sign Method and Projected Gradient Descent) to generate adversarial attacks for models. Can the authors use their analysis to identify submanifolds of adversarial examples?

The authors also show results only on the MNIST dataset, while these are promising, atleast extending their results to CIFAR10,100 will lend more support to their ideas. I also do not understand the difficulty in using fully trained models to perform their experiments. If there is an issue with a sparse G matrix then I suggest the authors try techniques like label smoothing to prevent this. If this cannot be done then that might suggest deeper issues with the paper, since the claim of describing the "model-centric data manifold" cannot be made if the fully trained model is not used.

While I have not studied the steps of Theorem 3.1 completely, it would help to add a step to equation 13 explaining how the Hessian came into the picture.

---

> ### Author Response · Authors · 2020-11-22
> **Answer to Reviewer 3**
>
> We answer according to the paragraphs of the review. Please see also our collective answer to the common issues raised by all Reviewers.
>
> -- Paragraph 2 and 3: We have enlarged our introduction to comment on generative models: they are not the main scope of this paper, however we have tried to elucidate the connection between the possibility of "navigating the data leaf" and generative models. Indeed we believe, as the Reviewer correctly suggests, that the main scope of our paper is to identify structure on data using a partially trained model. We have expressed this concept more in detail in the introduction and in sec. 3, 4 of our paper.
>
> -- Paragraph 4: we have tested our results also on CIFAR10 and included Appendix C for a visual report. We have also added a remark on "label smoothing" in Sec. 2.
>
> -- Paragraph 5: we have added some details to the proof of Thm 3.1, now in Appendix B, in particular we explained how the Hessian comes into the play, by giving explicitly the commutation formula for two vector fields.

---

### Official Review · AnonReviewer5 · 2020-11-04
**Proposing a method to analyze the input space of a deep neural network classifier.**

**Rating:** 5
**Confidence:** 3

**Review:**

Summary: The authors propose a so called data matrix that is induced in the input space of a deep neural network classifier. This matrix is similar to the Fisher-Rao metric, but for the input and not the parameters of the model. The analysis of this matrix shows that the classifier induces in the input space a specific structure, which the authors study. Constructive experiments are used in order to empirically verify the claims.

Comments:
1) In general, I find the paper a bit hard to follow and understand. In my opinion, the overall the writing of the paper can be improved. Some comments and suggestions that I believe will help:
- The proofs can be moved in the appendix to save space.
- The clarity of the paper can be improved. In general, be more explicit and provide clearer explanations.
- The motivation of the paper is not very clear to me. What is the "problem" that you aim to solve? Or what is the "purpose" of doing the proposed analysis. These goals should be more clear.
- The coherence of the paper can be improved.
- There are a lot of terms which are not defined explicitly. For instance, what is the foliation, the data leaf and its dimension?
- I suggest the authors to include some figures, which will help the reader to understand the proposed idea.
- I believe that the introduction is a bit unclear.

2) In general, after reading the paper, I am not sure if I can understand what is the main motivation? What is exactly the data leaves and the foliation?

3) I guess that the degeneracy of the matrix comes from the fact that, when the classifier is confident around a point only 1 class is selected and the probabilities do not change. So what is the purpose of under-training a model e.g. using 1 SGD step? I understand that this is done in order to avoid degenerate matrices, but why is interesting to study such a non-trained model?

4) How the submanifolds in the input space can be seen intuitively? I think an image will help a lot the understanding. Similarly, the distribution of Eq. 11 is not very clear.

5) I think that the proposed idea may be interesting. However, I believe that the current version of the paper needs to be improved such that to make the idea accessible. In my opinion, the current stage of the paper is not ready to be published.

===== After rebuttal =====

I appreciate the fact that the authors took into account our comments and improved their manuscript (+1), however, I think that there is still space for improvement.

My main concern, is the fact that we need the model to not be "fully trained" to do some analysis about "what the model learns". Therefore, regarding the data leaf, I am not sure if the trajectory that connects two points is actually moving on a "data manifold/leaf" or simply in the data space. Anyways, the behaviour of the metric away from the given data is kind of arbitrary since extrapolation analysis in neural networks is quite difficult. So the most crucial assumption is that the (learned) metric gives meaningful directions to move in the data space. In the experiments this seems to be the case "roughly", but the result could have been the same simply by the linear interpolation (which I think is missing). Similarly, if we just move linearly in a random direction, probably noise will appear gradually.

---

> ### Author Response · Authors · 2020-11-22
> **Answer to Reviewer 5**
>
> We answer here to the points raised by our anonymous Reviewer 5. We shall follow the numbering of the review.
>
> -- Answer to (1): we have followed the advices in order to improve clarity. In particular we have:
>
>    -- Moved involved proofs in Appendix B;
>
>    -- Provided more explanations especially in the mathematics part;
>
>    -- Enlarged our introduction so to elucidate better the scope of our paper and better show the coherence of its parts;
>
>    -- Defined more carefully mathematical terms (foliation, distribution etc) both in text and in the Appendix A;
>
>    -- Included more figures and a table;
>
>    -- Made an effort to make our introduction more understandable.
>
> -- Answers to (2): see answer to all Reviewers.
>
> -- Answer to (3): our partially trained model will indeed give the Riemannian structure to the data leaf. The model is trained for 1 _epoch_ and not 1 step. For MNIST such a model already reaches an accuracy over 90%. MNIST is particularly prone to overfitting, in CIFAR-10 experiments the model reach 90% accuracy, that is very close to best accuracy that can be obtained with VGG-11.
>
> -- Answer to (4): we added an image in Appendix A to explain how a foliation works in practice together with more explicit examples.
>
> -- Answer to (5): we take this opportunity to thank again the Reviewer, because we think he/she helped us greatly to explain our ideas and thus improve this manuscript.

---

### Author Response · Authors · 2020-11-22
**Comments to all Reviewers**

We would like to give an answer to the points
raised collectively by our Reviewers, that we wish to thank for their
careful reading of our work and their important remarks.
We shall adopt the convention AR1, AR3, AR4, AR5 to refer to our anonymous
Reviewers 1, 3, 4, 5.

-- Motivation of the paper. (AR5 comment 1, 2, AR3, AR1).

The purpose of our paper is to show that the data
matrix, defined in Sec. 3 using a partially
trained model, gives a manifold
structure to a portion of the data domain $\mathbf{R}^n$ containing our dataset.
We navigate this low dimensional
manifold, which we call the data leaf
with the experiments in Sec. 4. This will open the possibility
of defining local coordinates in the data leaf and to effectively
operate dimension reduction. We have elucidate
this point of view, together with the addition of the relevant literature
suggested by AR1, in the introduction and later on in Sec. 3 and 4.

-- Clarity of exposition and Mathematical background. (AR5 comment 1, 2, 4, AR1 comments 6, 7,
AR3 end of report)

Following the comment 1 of AR5 we have moved some proofs from Sec. 2, 3
to Appendix B. In this way the reading flows more naturally.
Also following the remarks of all of our Reviewers,
we have added more explanation to our mathematical
treatment: we expanded Sec. 3 and App. A adding some non
trivial geometrical examples a reader can visualize and compute
relatively quickly. This is also in response
to comment 2 of AR5.
We also now introduce our mathematical
results highlighting the consequences for the empirical
implementations following AR1 suggestions.

-- Partially trained model. (AR5 comment 3, 4, AR1 comments 3, 5)

A fully trained model will give a low rank
local data matrix $G(x,w)$, thus
unable to discern the data manifold structure
that we aim for. We have added some details to our explanation in Sec. 3 and 4 to elucidate this phenomenon.

-- MNIST Dataset.  (AR1, comment 5, AR3)
We have added some remarks on the Cifar10 Dataset as
suggested by our Reviewers and Appendix C with some
relevant paths in the data leaf for the Cifar10 Dataset.

---

### Decision · Program_Chairs · 2021-01-07
**Final Decision**

**Decision:**

Reject

**Comment:**

The paper defines a "local data matrix" (inspired from local Fisher matrix) and uses it to obtain a foliation in the data space. This provides a lens to view the data space from model's perspective. While the idea is interesting, reviewers have two main concerns from the reviewers which are not fully addressed in the author response:
(i) The method works with partially trained model (1 epoch for MNIST) and it's not clear how the observations made in the paper extend to fully trained models,
(ii) The motivation and application of the proposed model-centric view of data space needs more work - it will be good to think of some applications where this view can help.

I encourage the authors to consider the suggestions from the reviewers (e.g, R3 suggested label smoothing for (i)), and submit a revised version to a future venue.